# Artifact Management for Cerebral Near-Infrared Spectroscopy Signals: A Systematic Scoping Review

**DOI:** 10.3390/bioengineering11090933

**Published:** 2024-09-18

**Authors:** Tobias Bergmann, Nuray Vakitbilir, Alwyn Gomez, Abrar Islam, Kevin Y. Stein, Amanjyot Singh Sainbhi, Logan Froese, Frederick A. Zeiler

**Affiliations:** 1Biomedical Engineering, Faculty of Engineering, University of Manitoba, Winnipeg, MB R3T 5V6, Canada; vakitbir@myumanitoba.ca (N.V.); islama9@myumanitoba.ca (A.I.); steink34@myumanitoba.ca (K.Y.S.); amanjyot.s.sainbhi@gmail.com (A.S.S.); 2Section of Neurosurgery, Department of Surgery, Rady Faculty of Health Sciences, University of Manitoba, Winnipeg, MB R3A 1R9, Canada; gomeza35@myumanitoba.ca; 3Department of Human Anatomy and Cell Science, Rady Faculty of Health Sciences, University of Manitoba, Winnipeg, MB R3E 0J9, Canada; 4Undergraduate Medicine, Rady Faculty of Health Sciences, University of Manitoba, Winnipeg, MB R3E 3P5, Canada; 5Department of Clinical Neuroscience, Karolinska Institutet, 171 77 Stockholm, Sweden; log.froese@gmail.com; 6Centre on Aging, University of Manitoba, Winnipeg, MB R3T 2N2, Canada; 7Division of Anaesthesia, Department of Medicine, Addenbrooke’s Hospital, University of Cambridge, Cambridge CB2 0QQ, UK; 8Pan Am Clinic Foundation, Winnipeg, MB R3M 3E4, Canada

**Keywords:** artifact management, cerebral near-infrared spectroscopy, cerebral hemodynamic monitoring, bio signal analysis

## Abstract

Artifacts induced during patient monitoring are a main limitation for near-infrared spectroscopy (NIRS) as a non-invasive method of cerebral hemodynamic monitoring. There currently does not exist a robust “gold-standard” method for artifact management for these signals. The objective of this review is to comprehensively examine the literature on existing artifact management methods for cerebral NIRS signals recorded in animals and humans. A search of five databases was conducted based on the Preferred Reporting Items for Systematic Reviews and Meta-Analysis guidelines. The search yielded 806 unique results. There were 19 articles from these results that were included in this review based on the inclusion/exclusion criteria. There were an additional 36 articles identified in the references of select articles that were also included. The methods outlined in these articles were grouped under two major categories: (1) motion and other disconnection artifact removal methods; (2) data quality improvement and physiological/other noise artifact filtering methods. These were sub-categorized by method type. It proved difficult to quantitatively compare the methods due to the heterogeneity of the effectiveness metrics and definitions of artifacts. The limitations evident in the existing literature justify the need for more comprehensive comparisons of artifact management. This review provides insights into the available methods for artifact management in cerebral NIRS and justification for a homogenous method to quantify the effectiveness of artifact management methods. This builds upon the work of two existing reviews that have been conducted on this topic; however, the scope is extended to all artifact types and all NIRS recording types. Future work by our lab in cerebral NIRS artifact management will lie in a layered artifact management method that will employ different techniques covered in this review (including dynamic thresholding, autoregressive-based methods, and wavelet-based methods) amongst others to remove varying artifact types.

## 1. Introduction

Near-infrared spectroscopy (NIRS) is a method of measuring underlying hemodynamics within the brain. It can be used to measure cerebral autoregulation (CA) in a clinical setting non-invasively, having far-reaching impacts. However, the widespread clinical potential of this technology has been hindered by artifacts contaminating the signal, as there are no robust artifact management methods available.

NIRS applied as a method of measuring regional cerebral tissue oxygenation (rSO_2_) was proposed initially by Frans F. Jöbsis in 1977 [1]. The calculation of rSO_2_ leverages the Modified Beer–Lambert Law, which describes the absorption of light as it passes through different media [2]. Specific wavelengths within the near-infrared range correspond to the absorption peaks of deoxyhemoglobin and oxyhemoglobin, respectively. As such, NIRS can be used to measure their respective changes in concentration by measuring the absorption of certain emitted wavelengths [3,4].

The use of rSO_2_ and other related metrics (optical density, concentrations of oxyhemoglobin (HbO), and concentrations of deoxyhemoglobin (HHb)) in a clinical setting has grown in popularity, as it is a non-invasive surrogate method for measuring cerebral blood flow (CBF) [5]. CBF is critical in the evaluation of patient cerebral autoregulation (CA), which is the ability of the brain to maintain a constant CBF despite changes in cerebral perfusion pressure (CPP) [6]. Impaired CA can result from traumatic brain injury [7,8,9,10], stroke [11], meningitis [12], and cardiac arrest [13] and can put the patient at risk of secondary injuries from cerebral tissue hypoxia and edema [7].

Artifact-free data facilitate improvements to clinical treatment in instances of impaired CA, as more timely identification, intervention, and prevention of secondary injuries becomes possible with accurate data. To research into the understanding of high temporal relationships within cerebral physiology, current research is focused on multivariate relationships within cerebral physiology to derive new multi-modal time-series metrics. The main concern for the widespread clinical implementation of NIRS-based metrics is that monitoring conducted at the bedside is often riddled with artifacts from device disruption, patient motion, and loss of signal quality [14,15,16,17]. Signals can also be contaminated by physiological or outside noise. The intricate and varied morphological features of artifacts observed in NIRS signals undermine the efficacy of simple filtering methods, such as thresholding and gross mean value assessment, in this application. Consequently, manual signal clearing must be conducted. Manual signal clearing requires trained personnel, is costly, has the potential for human error, and is time consuming, all of which impede the timely use of these data for clinical applications [18]. As such, there is an on-going search for a broadly accepted semi or fully autonomous method of artifact clearing cerebral NIRS signals in real time [16,18]. A systematically conducted scoping review of current semi and fully autonomous methods of artifact management will help elucidate the types of methods that have been developed and their respective efficacies. The information presented in this review will help build towards a “gold-standard” artifact management method for cerebral NIRS signals.

The objectives of this systematically conducted scoping review include providing an overview of artifact management methods that have been developed specifically for cerebral NIRS signals, identifying if a leading method has already been developed, and investigating how this could be accomplished. The breadth of artifact removal techniques, including accelerometer-based methods, wavelet-based methods, machine learning-based methods, filter-based methods, component analysis-based methods, and hybrid methods, amongst others that did not fit into the previous categories, were explored. Additionally, methods to remove signal drift and signal noise were also examined. The main goal of this work is to categorize and evaluate existing methods of artifact management for cerebral NIRS to determine whether a leading method has already been identified. Additionally, it sought to examine which artifact management techniques could be used as the basis for future research. The analysis conducted encapsulates the current state of knowledge regarding artifact management techniques for cerebral NIRS signals, respective success rates in artifact removal, the limitations prevalent in the literature, and implications for future research.

Future work by this lab will involve the construction of a sequential layered artifact management pipeline. This model will aim to characterize artifacts in cerebral pressure-flow physiologic signals, including cerebral NIRS, using a combination of signal analysis methods. As such, the motivation for conducting this review is to understand current methodologies to use as a basis for the development of this future algorithm.

Section 2 details the Methodology that was followed in conducting this systematic review, including the search question that was posed, the development of search criteria, and how data collection was conducted. Section 3 describes the results of the systematic review as the methods extracted from the databases were organized into sub-categories and their respective efficacies were compared. Also in this section was an overview of some of the more robust comparisons that have been conducted on the efficacy of different artifact management methods. Section 4 discusses the findings of this review as well as provides limitations to the literature, limitations to the review itself, and future directions for work in this domain. Finally, Section 5 provides a conclusion to the manuscript. Appendix B includes the PRISMA-ScR checklist, Appendix C displays the search string that was entered into the databases, and Appendix A provide the data that were extracted from each article included in this review.

## 2. Materials and Methods

This systematically conducted scoping review followed the methodology outlined in the Cochrane Handbook for Systematic Reviews [19]. The reporting of the results conforms to the Preferred Reporting Items for Systematic Reviews and Meta-Analysis (PRISMA) guidelines with the PRISMA Extension for Scoping Review [20,21]. The review objectives for the search strategy were developed collaboratively by TB and FAZ, with help in the article filtering process by NV (PhD Student) and LF. The completed PRISMA checklist can be found in Appendix B (Table A1)

### 2.1. Ethical Considerations

All articles examined in this review were from previously published journals and are expected to have been vetted by these journals. Thus, specific ethics approval for this systematically conducted scoping review was not required.

### 2.2. Search Question and Inclusion/Exclusion Criteria

The question examined within this review was “what methods have been used to manage artifacts in continuous cerebral NIRS signal sources?” Additional questions included the signal types for which artifact management methods were developed as well as the efficacies of the methods in removing artifacts, particularly compared to other methods.

Continuous cerebral near-infrared spectroscopy (NIRS) signals were defined as including oxygen saturation (SpO_2_), rSO_2_, concentration of oxyhemoglobin (HbO), concentration of deoxyhemoglobin (HHb), optical density, and related measures that were recorded at a minimum of 1 Hz. All articles that included experimental animal or human data on artifact management of raw or processed continuous NIRS signals were included. Articles were required to be full-length and published in English. The following articles were excluded from the analysis: those published in languages other than English, abstracts only, theoretical or simulation studies, non-time series data, and artifact management for non-cerebral NIRS signals.

### 2.3. Search Strategy

Searches were conducted across multiple databases including BIOSIS, SCOPUS, EMBASE, PubMed, and Cochrane Library, covering the entire period from the conception of each database up to 26 January 2024. As such, it represents the state of literature on this topic up to this date. Dedicated search strings were constructed for each, consisting of terms/synonyms for NIRS and artifact methods. The detailed search strategy for each database is provided in Appendix C. The results of the search from each database were compiled and deduplication was conducted.

### 2.4. Study Selections

A manual review of all remaining articles after deduplication from the initial search was conducted using a two-stage two-reviewer approach. During the first stage, two reviewers (TB and NV) independently screened the title and abstract of each article for inclusion/exclusion criteria. The included studies were then screened a second time by both reviewers, this time evaluating the entire content of each article for inclusion and exclusion criteria. Any disagreements between the two reviewers were resolved by a third-party (LF and FAZ).

### 2.5. Data Collection

Characteristics of each study were recorded for analysis including the main purpose of the study and specific patient/subject and data information. Patient/subject information included the sample size, sex, history of neurological injury, and methodology signal recording. Data information included the sampling rate of the NIRS signals, type of recording system used, the anatomical location and number of NIRS optodes, and any data simultaneously being recorded. Information regarding the details of methods described in each article for the removal artifacts from the cerebral NIRS signal were also extracted. Analysis of each method involved the evaluation of the effectiveness and main study results. Limitations in the effectiveness or ease of implementation for each method were also extracted, including those identified by the authors and those prevalent during the critical analysis of each article.

The data collected are summarized in Appendix A. Note that this was a review of all available manuscripts. Please refer to the original publications for more details about data availability.

## 3. Results

The results of the search strategy that was conducted across all databases and the reference sections of each article are depicted in a PRISMA flow diagram in Figure 1. There were 1440 articles identified based on the search strings utilized across all databases searched. A total of 634 were removed as they were duplicates of other references, and the remaining 806 entered the first of the two-stage reviewing process.

Through the application of inclusion and exclusion criteria to the title and abstract, 763 were removed, leaving 43 articles to enter the second stage. The portable document format (PDFs) of each article was obtained. Applying the inclusion and exclusion criteria to the full-length PDFs resulted in 24 articles being excluded and 19 being included. An additional 36 articles were included from the reference section of the pertinent articles, and there were several of these due to the different ways these articles were indexed in various databases. All 55 included studies were human-based studies, and there were no animal-based studies that fit all the inclusion criteria. There were several different NIRS and fNIRS systems that were used to record cerebral NIRS signals. Some of these systems were commercially available. The most commonly used of these included the Hitachi ETG-4000 [22,23,24,25,26,27] and the NIRSOptix TechEn CW6 [28,29,30,31,32,33,34,35]. Other systems included those that were custom-made [36,37,38,39,40,41,42,43,44,45,46] or presented in another academic work [47,48]. Each system derived their respective cerebral NIRS-based metric using a different proprietary algorithm; however, each of these methods fundamentally assess the percent saturation or concentration of hemoglobin in cerebral blood flow. Therefore, despite the potential differences in the methodology used to derive these measures, the fundamental pathophysiology being described is similar. Thus, the artifact management methods included in this review were assumed to be sufficiently functional across these measures to warrant the common consideration.

There were two main categories of artifact removal methods yielded from the database search: (1) motion and other disconnection artifact removal methods; (2) signal quality improvement and physiological/other noise removal methods. These differ in that motion and other disconnection artifacts are morphological errors—for example, large spikes in magnitude or signals being lost. In contrast, physiological and other noise artifacts contaminate signals with noise such that they are not useable as it becomes difficult to discern physiological signals for the quantification of CBF from other noise in the signal.

### 3.1. Motion and Other Disconnection Artifact Removal Methods

There were 40 articles that outlined motion and other disconnection artifact management methods. The artifact management methods outlined were all developed for NIRS or fNIRS signals measured with a sampling rate of 1 Hz to 1000 Hz, apart from a source which developed artifact management for near-infrared imaging (NIRI) signals. There were 26 of 40 articles that outlined methods developed for HbO and HHb signals, five methods for HbO only, one method for rSO_2_, four methods for optical density signals, and four methods for an unstated NIRS signal. Based on the main method used for artifact removal, the motion and other disconnection methods were grouped into six categories: accelerometer-based methods, wavelet-based methods, machine learning-based methods, filter based-methods, component analysis-based methods, hybrid methods, and other methods. Each method is summarized below and details regarding the populations on which they were tested, the efficacy, results and limitations of each method are provided in Appendix A with a summary of Appendix A provided in Table 1.

#### 3.1.1. Accelerometer-Based Methods

There were nine motion and other disconnection artifact removal methods that leveraged accelerometer data that was simultaneously recorded with NIRS or fNIRS signals [31,36,42,43,47,49,50,51,53]. These methods use thresholds in the accelerometer data to detect when the patient or the NIRS probe was moved during recording. When movement is recorded, the algorithms identify the segment as artifactual. Many of the studies compared a proposed method to only one or two other methods, with little consistency between studies with respect to methodological approaches such as sampling rate, demographics of patient population, protocols during signal recording, and types of artifacts being removed. The availability of an accelerometer for simultaneous measurement with cerebral NIRS will affect whether these artifact management methods can be extended to all recording setups in clinics.

#### 3.1.2. Wavelet-Based Methods

Wavelet analysis involves transforming time-series NIRS signals into the time-frequency domain. Wavelet coefficients are based off the frequency of the oscillations of the NIRS signal at a particular time. Wavelet-based thresholding was used to disseminate between physiological and erroneous signals. Continuous and discrete wavelet-based methods are used in eight of the identified artifact removal methods [17,22,23,55,56,57,58,59]. It was difficult to compare the effectiveness of each method due to the considerable heterogeneity in the effectiveness metrics (signal-to-noise ratio (SNR), percent artifact reduction, mean squared error (MSE), normalized MSE, beta values, t-statistic, coefficient of determination (R^2^)) data sampling rates, different patient populations used (newborn, healthy adult, traumatic brain injury, elective spinal), and different variations in artifacts (simple signal loss, magnitude spikes, or more complex morphological artifacts).

#### 3.1.3. Machine Learning-Based Methods

There were three methods that leveraged machine learning techniques for artifact identification and removal [37,62,64]. Russell-Buckland et al. proposed a method that used feature engineering to identify artifacts based on power density fraction, sample entropy, autocorrelation, and area under the curve for the signal based on artifact-free signals. The data was then fed into a Random Forest algorithm. However, there was limited success in artifact identification based on the engineering feature selection. Kim et al. developed a deep convolutional neural network that extracted features from fNIRS signals and used weighted filtering to reconstruct the hemodynamic response function without motion artifacts [62]. This method was able to extract the synthetic hemodynamic response function better than a wavelet-based method [61] and an autoregressive-based [60] method using metrics of MSE and performance activation detection measured using AUC-ROC [62]. Lee et al. proposed a method that used a 10-band wavelet transformation that is fed into a back propagation neural network (BPNN) to remove motion artifacts from fNIRS signals [64]. The resulting CNR indicated that this method was able to perform denoising and global detrending as it outperformed two wavelet-based methods and another HRF smoothing method [64].

#### 3.1.4. Filter-Based Methods

There were six artifact removal methods that were filter-based. Two of these methods were developed by Izzetoglu et al. These methods use Wiener filtering and Kalman filtering, respectively [38,39]. These two methods were compared and showed similar success in artifact removal based on SNR. They both had better SNR results than an adaptive filter for HHb and HbO signals recorded from 11 healthy volunteers [39] The average improvement of the SNR (ΔSNR) for the adaptive, Wiener, and Kalman filters were 3.4396, 6.6980, and 7.6548, respectively, when the filters were employed to remove artifacts originating from head movements [39]. There was another method that extended the Kalman filtering method proposed by Izzetoglu et al. [39] using an autoregressive moving average (ARMA) model to translate the signal into the state space prior to applying the Kalman filter [44]. This extension increased the fNIRS ΔSNR from 8.7 dB (using the Kalman filter [39]) to 10.4 dB [44]. Dong and Jeong also proposed the use of an extended Kalman filter-based method [45]. It outperformed the linear Kalman filtering and adaptive filtering techniques presented by Zhang et al. [72] when tested on a dataset previously published by this author [73]. A synthetic HRF was able to be more accurately extracted in the presence of experimentally measured artifacts by metrics of root mean squared error (RMSE), percent root difference (PRD), and correlation coefficients [45]. Huang et al. described a dual-stage median filter capable of removing step-like and spike artifacts from fNIRS signals, outperforming wavelet-based [23] and spline interpolation-based [41] methods by metrics of signal distortion ratio (SDR) and normalized mean squared error (NMSE) [66]. Robertson et al. also developed a method that utilized recursive least squares for an adaptive filter; however, this method had limited success compared to ICA-based, wavelet-based, and regression-based methods determined using SNR [17].

#### 3.1.5. Component Analysis-Based Methods

There were four artifact management methods that leveraged component analysis. Yanhua Shi et al. developed an independent component analysis (ICA)-based method that demonstrated a mean ΔSNR for the HHb and HbO signals of 2.336 and 2.139, respectively, for the raw datasets recorded from four healthy adults (mean age of 23) [67]. The ΔSNR using the ICA method was two times greater than the ΔSNR noted in the compared wavelet-based method, which had mean ΔSNRs of 1.191 and 1.118 in HHb and HbO, respectively [67]. There was also a targeted principal component analysis (tPCA) method developed. It was compared to a spline-based method and a wavelet-based method [28]. tPCA had a better performance than both methods (spline-based and wavelet-based) in correlation between the cleaned signal and the true HRF. However, tPCA performed worse in mean squared error than the spline-based method when tested on datasets recorded from five healthy adults (ages ranging from 23 to 52) [28]. There was an accelerometer-based method presented by von Lühmann et al. [53]. This method used an independent component analysis by an entropy rate bound minimization (ICA-ERBM) algorithm to decompose fNIRS signals before temporally embedding accelerometer data. It then identified artifacts using shared components detected using canonical component analysis (CCA) and estimated the artifact length in the ERBM space. This algorithm improved the SNR of continuous hemodynamic signals up to 10 dB and reduced motion artifacts by an order of two, outperforming several conventional methods in extracting the HRF [53]. However, future work is needed. Robertson et al. presented an ICA-based method that was able to achieve an average increase in SNR for NIRS signals of 5.62 dB and 2.76 dB when λ = 760 nm and λ = 830 nm, respectively on a dataset of three subjects where the temporal location of the artifacts was known. This method also achieved the best performance in SNR (3.20 dB, 3.67 dB) when the temporal location of the artifacts was unknown (compared to a wavelet-based, multi-channel regression-based and adaptive filter-based method) [17].

#### 3.1.6. Hybrid Methods

Five hybrid artifact removal methods were identified. The first was an approach that uses dynamic thresholding and circumstantially employs different methods based on the type of artifact. Large oscillations are corrected using cubic spline interpolation, baseline shifts are corrected using spline interpolation, and slight oscillations are corrected using a two thresholds in a wavelet-based method [40]. Jahani et al. presented another hybrid artifact clearance method that utilizes a low-pass and Sobel filter to identify motion artifacts based on signal gradient values deviating from typical physiological variations, amplitude thresholding to identify baseline shifts, spline interpolation to model the motion artifact epoch, and Savitzky–Golay filtering for the remaining high-frequency artifact removal [32]. An in-depth analysis of this method is included in Section 3.1.8. The final hybrid method presented is a movement artifact reduction algorithm (MARA) that includes six steps using moving standard deviation (to detect motion artifacts) and spline interpolation (to correct artifacts) [41]. PRD, RMSE, and Pearson’s correlation coefficient (R) were calculated between the clean signal and the signal with induced motion artifacts (MA) as well as between the clean signal and the signal with induced MA that was cleaned using MARA. Employing proposed MARA on three datasets of an undisclosed size and demographic resulted in an average change of −89.7% in PRD, −89.8% in RSME, and an increase of 61.6% in R when induced motion artifacts were removed [41]. Gu et al. proposed a hybrid method that used thresholding for artifact detection, empirical mode decomposition for artifact removal, and spline interpolation to maintain continuity [27]. The use of this method resulted in an average increase in SNR of 53%, reduction in average MSE of 47%, and an R^2^ between the processed and true signal of 0.79. This method generally outperformed the spline interpolation-based method [41], wavelet-based method [23], and kurtosis wavelet-based method [56]. Zhou et al. presented a hybrid method that first used a moving standard deviation (MSD) filter to detect the onset of artifacts in a contaminated fNIRS dataset, then used cubic spline interpolation to remove them, then used a smaller MSD filter to detect more subtle artifacts, and used Savtitzky–Golay (SG) filtering to denoise these signals [46]. The proposed method was able to achieve a SNR of 2.41 dB whereas the use of only spline interpolation or SG filtering resulted in SNR values of −19.96 dB and −23.79 dB, respectively.

#### 3.1.7. Other Methods

There were nine methods that did not fall under the previous categories. Barker et al. developed a method using an adjusted autoregressive model with a pre-whitening filter and iteratively reweighted least squares (AR(P)-IRLS) [60]. This method outperformed an ordinary least squares regression method, a wavelet-based method with ordinary least squares regression, and a spline-based method with ordinary least squares regression using receiver operating characteristic (ROC) analysis for a synthetic dataset and a dataset of 22 children [60]. The ROC is a plot developed based on sensitivity and specificity to evaluate the successfulness of the noise removal algorithm. The AR(P)-IRLS method was extended using a dual-stage Kalman filter that can be applied in real time (online application) and performs similarly [34]. A correlation-based signal improvement method was developed by Cui et al. which was able to improve the contrast-to-noise ratio (CNR) from 1.31 and 1.28 to 2.59 and 2.59 for HHb and HbO signals, respectively, tested on a dataset of 10 healthy adults [24]. The Temporal Derivative Distribution Repair (TDDR) method was developed by Fishburn et al. [33]. This method involves taking the temporal derivative of the cerebral NIRS signal, initializing a vector of observation weights, iteratively estimating the robust observation weights, applying the resulting robust weights to the centered temporal derivative to produce the corrected derivative, and integrating the corrected temporal derivative to yield the corrected signal [33]. An in-depth evaluation of this method is included in Section 3.1.8. Wang and Seghouane developed a method using discrete cosine transformation coefficients to estimate the signal in the presence of artifacts [68]. This was tested on HbO signals with artifacts introduced measured from an undisclosed number of children. The ability of this method to remove artifacts was compared to two targeted artifact removal algorithms (TARA (convex) and TARA (non-convex)). TARA algorithms assume a model of the HbO signal, a low-pass signal, two types of artifacts (step-discontinuities and signal magnitude spikes) and a Gaussian stochastic process. Convexity refers to the number of minima used for optimization [74]. These algorithms estimate the components of each portion of the signal model using optimization to remove the identified artifacts and reconstruct the hemodynamic signal [74]. The proposed algorithm presented by Wang and Seghouane directly estimates the signal parameters, while the TARA algorithms do so indirectly. The mean squared error (MSE) between true resting state signal and restructured signal were 0.020876 for TARA (convex), 0.015514 for TARA (non-convex), and 0.0037094 for the proposed method [68]. Another method was the functions that were included in the NICA toolbox for NIRS calculations and analyses. These included a common average reference method, transfer function models as well as the low pass Butterworth filter and Grand Average and Region of Interest Analysis [69]. Sutoko et al. presented a method that used sudden detected increases in fNIRS signal amplitude, a shift in baseline amplitude, and intertrial discrepancy of correlation as indicators of the presence of artifactual segments [70]. This simple algorithm was able to achieve a rejection accuracy of 71.8% when compared to visual inspection. Robertson et al. developed a multi-channel regression method that used linear regression between 30 channels of NIRS data to improve its SNR. This method performed similarly to the compared ICA-based method and outperformed a recursive least squares-based adaptive filter and a wavelet-based method [17]. Sweeney et al. presented an EEMD-CCA method that was able to reduce motion artifacts from fNIRS signals, achieving a ΔSNR of 3.5 dB, a 49.4% artifact reduction, and a correlation with the ground truth signal of 0.68 [35]. This method outperformed an EEMD-ICA [71] as well as the wavelet-based method presented by Robertson et al. [17,35]

#### 3.1.8. Comparison of Methods

It proved difficult to compare different artifact removal methods due to the heterogeneity of the datasets based on factors like demographics, different variations of artifacts, and different metrics of measuring effectiveness, as was alluded to in Section 3.1.2. However, there were four studies that included more in-depth comparisons to other artifact removal methods in NIRS signals.

The first of these studies was the evaluation of a hybrid motion artifact detection and removal method [40]. The SNR and R values between the processed and clean signal was used to demonstrate the effectiveness of the proposed method for NIRS data at 10 Hz (40 patients, mean age 32 years) in comparison to a wavelet-based method, accelerometer-based method, median filtering, spline interpolation with Savitzky–Golay filtering (Spline-SG), spline interpolation with robust locally weighted scatterplot smoothing (Spline-Rloess), and solely cubic spline interpolation combined with wavelet filtering. The proposed method was the only method in the performance comparison between methods with an SNR that exceeded 0 dB and a value of R above 0.6 between the clean and processed signals [40].

There was a second study comparing several methods to a developed hybrid artifact clearance method [32]. This method was validated against other techniques presented in articles included in this review including wavelet filtering [41], tPCA [28], and correlation-based signal improvement (CBSI) [24] as well as combinations of more broad signal techniques like Rloess for artifact spike removal and spline interpolation for baseline shifts. These methods, and combinations of these methods, were applied to the same datasets as the proposed methods. The evaluation of the methods was based on the ability of each method to recover the synthetic hemodynamic response function that was applied to two raw NIRS datasets (seven and five patients, respectively) recorded at 50 Hz (one set performing actions and the second at rest) with added baseline shift and artifacts [32]. The signal processed by each signal was compared to the true HRF by metrics of MSE, peak-to-peak error (E_p_), coefficient of determination (R^2^), and area under the receiver operator characteristic (AUC-ROC). For the first dataset, a combined method of Rloess and spline interpolation (Rloess-Spline) had the best performance in metrics MSE (0.60 ± 0.16 × 10^4^), E_p_ (3.90 ± 1.13 × 10^4^), and R^2^ (0.80 ± 0.02), and CSBI had the best performance in AUC-ROC (0.91 ± 0.03). For the second dataset, the proposed Spline-SG method had the best performance in metrics MSE (0.44 ± 0.06 × 10^4^) and E_p_ (2.52 ± 0.41 × 10^4^), CSBI had the best performance in R^2^ (0.84 ± 0.01), and the spline only method, the proposed Spline-SG, and Rloess-Spline method had identical AUC-ROC results (0.89 ± 0.05).

A third study compared the TDDR method [33] to several other methods including CBSI [24], movement artifact reduction algorithm (MARA) [41], tPCA [28,75], kurtosis wavelet filtering (kWavelet) [23,56], and the previously mentioned Spline-SG method [32]. These methods were evaluated using simulated NIRS data as well as experimental NIRS data (23 patients, ages 7 to 15, conducting working memory tasks) recorded at 50 Hz and resampled at 5 Hz after pre-processing using the NIRS Brain AnalyzIR toolbox [76]. The success of the artifact removal methods for the simulated data was evaluated using AUC-ROC compared to the motion-free data. The rankings of the magnitude of AUC-ROC values compared to the motion-free data (0.869) were as follows: TDDR (0.775), CSBI (0.733), Spline-SG (0.652), tPCA (0.591), MARA (0.563), uncorrected data (0.516), and kWavelet (0.513). The success of the artifact removal for the experimental data was evaluated using the maximum activation t-statistic and greatest number of mesh vertices with positive significant (*p* < 0.05) values. The rankings of the magnitude for the t-statistic were as follows: TDDR (4.88), kWavelet (3.96), tPCA (3.79), Spline-SG (3.68), uncorrected (3.67), CSBI (3.02), and MARA (2.96). The rankings of the magnitude for mesh vertices with positive significant values were as follows: TDDR (2399), uncorrected (1560), Spline-SG (1153), kWavelet (935), MARA (924), CBSI (903), and tPCA (891). The TDDR method outperforms all other methods in simulated and experimental data; however, limitations to the method are prevalent. High-frequency oscillations increase the variance of the temporal derivative, which results in the TDDR method not performing well for high-frequency artifacts [33]. The Spline-SG method indicated strong results in the previous two articles outlined; however, it performs poorly in achieving peak activation [45,46,49]. Activation is a metric to analyze the abilities of each model to detect changes in the HbO signals in response to tasks in the brain. It is evaluated using a general linear model (GLM) [77]. Poor performance in peak activation implies that the algorithm was not able to detect these working load changes. CSBI performed well in the simulated data, but performed poorly in the experimental data due to its reliance on the assumption that there is a strong anti-correlation between HHb and HbO [48,49].

A final study that discussed the performance of five different methods in motion artifact removal was presented by Robertson et al. [17]. The methods compared included two multiple channel regression-based methods (a 2-channel and a 30-channel), a discrete wavelet-based method, a recursive least squares-based adaptive filter, and an ICA-based method. These methods were tested on two datasets, the first when motion was induced when the subject was instructed to move their head (three subjects) and another when the subject was told to tap their finger (one subject). When time of motion was known (set of three subjects) the average across the three subjects for SNR (dB) when λ = 760 nm was wavelet (5.95), 30-channel regression (5.67) and ICA (5.62) and when λ = 830 nm, the ranking was the same: wavelet (4.93), 30-channel regression, and (4.56) ICA (2.75). However, using the subject dataset when the motion was not known, the ranking changed. When λ = 760 nm, ICA (3.20 dB), 30-channel regression (3.01 dB), and wavelet (0.89 dB) and when λ = 830 nm, ICA (3.67 dB), 30-channel regression (3.01 dB), and wavelet (0.58 dB) [17]. This change in ranking indicates that the wavelet-based method may not perform when the location of motion is not known. However, due to the small dataset, more work is needed. It was able to conclude based on their performance that the RLS method and 2-channel regression method had substantially worse performances than the rest [17].

### 3.2. Data Quality Improvement and Physiological/Other Noise Artifact Filtering Methods

There were 15 articles presenting methods that were devised for data quality improvement and physiological/other noise artifacts. The methods were developed for NIRS signals with sampling rates ranging from 1.81 Hz to 100 Hz. There were 9 of 13 methods developed for HHb and HbO signals, 1 method with HbO, and 1 method for an undisclosed NIRS signal. These methods were organized into three categories: signal drift removal methods, physiological/other noise artifact removal methods using NIRS only, and physiological/other noise artifact removal methods using auxiliary signals only. More detailed information on the studies that evaluated or incorporated these methods (patient information, data information, methodology, effectiveness, results, and limitations) are provided in Appendix A with a summary of Appendix A provided in Table 2.

#### 3.2.1. Signal Drift Removal Methods

There were two articles that presented a method developed solely to remove signal drift in cerebral NIRS signals utilizing techniques such as removal of estimated induced functional response using wavelet coefficient thresholding [25]. This method increased the CNR of the HbO and HHb signals from below 1 (in raw data) to above 6 in HbO and above 5 in HHb signals for the five different channels analyzed. This method outperformed the wavelet-minimum description length (wavelet-MDL) method [89] to which it was compared. This method increased the CNR in HbO and HHb signals to below 3 for either channel [25]. However, this method did nothing to remove any other noise. This method was extended to estimate HRF in the presence of noise and drift [78]. It outperformed two established methods using experimental NIRS data.

#### 3.2.2. Physiological and Other Noise Artifact Removal Methods—NIRS Only

There were eight methods developed to remove extraneous physiological and other noise using a variety of techniques. Nguyen et al. developed a recursive least squares-based method that yielded a noise reduction in HbO and HHb signals of 77% and 99%, respectively, based on CNR data [80]. This performance was better than the Kalman filter method, low-pass filter method, and independent component analysis (ICA)-based method. Zhang et al. developed a multitiered adaptive filter method to remove drift that improved the CNR compared to the raw HbO signal from 40.2% to 80.8% [72] for data recorded from 1 healthy volunteer and 64% to 75% [81] for data recorded from 17 healthy volunteers. However, this method decreased the CNR for HHb before and after filtering [72,81]. Additional methods for this purpose include a combination of low-pass filter, mean average high-pass filter, targeted canonical component analysis (tCCA), Kalman filter [29], a three-tiered approach involving empirical mode decomposition, independent component analysis, and correntropy spectral density [83], an additional ICA-based method [82], and a comparison of several methods of pre-processing and regression for physiological noise removal [84]. Santosa et al. indicated that an iterative auto-regressive least squares (AR-IRLS) with a short separation method performed better than PCA and the ordinary least squares methods [84]. Only the AR-IRLS algorithm evaluated had an AUC-ROC value above 0.75 [84]. Guerrero-Mosquera presented a study comparing running correlation (global information) to identify noisy artifact ridden fNIRS channels to using cross-correlation (local information) [85]. Under different cognitive conditions (2-back and 0-back tasks) the AUROC in identifying noisy channels was 60.57% using running correlation (global correlations) and 91.23% using cross correlations (local correlations) [85].

#### 3.2.3. Physiological and Other Noise Artifact Removal Methods—Auxiliary Signals Used

There were five methods developed that use simultaneously recorded auxiliary signals to identify physiological noise in cerebral NIRS signals. Auxiliary data sources included blood pressure coupled with photoplethysmography (PPG) and accelerometer data, blood pressure coupled with electrocardiograms (ECGs), respiration rate data blood pressure coupled with respiration rate, functional magnetic resonance imaging (fMRI) data, and skin blood flow measured by laser Doppler [26,30,86,87,88]. The auxiliary signals are used to identify segments where artifacts occurred by comparing whether oscillations in cerebral NIRS signals are of a particular hemodynamic physiological origin or are erroneous. It was difficult to conduct a quantitative comparison of the effectiveness of these methods due to the heterogeneity of metrics that were used and differences in what was defined as “unwanted noise” that required removal. The method developed by Bauernfeind et al. demonstrated that the common average reference (CAR), transfer function (TF), and ICA methods resulted in increases of CNR in HbO signals using simultaneously measured BP, respiration rate, and ECG signals using the TF method. There was an improvement of CNR between 3.7% to 188.8% in HbO signals across datasets. The improvement of CNR was between −0.6% to 55.0% in HHb signals across datasets. The TF method demonstrated the best results of the three methods [26]. The TF was the only method that obtained any HHb signal improvement. Bontrager et al. indicated that there was greater improvement at reducing the correlation between fNIRS and BP signals using an mutual information filter compared to the raw data and RLS method [86]. Another study compared the ShortPCA GLM method to the standard GLM, MS-ICA method (proposed by Molgedey and Schuster [90]), and RestEV method (eigenvector-based spatial filtering method [75]). The ShortPCA GLM method used simultaneously recorded fMRI signals and was determined most appropriate for fitting changes in oxyhemoglobin during movements [88]. Finally, the GLM with short separation (SS) depicted superior results based on the yielded robustness of HRF estimation compared to GLM with tCCA [30].

## 4. Discussion

The aim of this systematically conducted scoping review of artifact management for cerebral NIRS signals is to provide a comprehensive overview of existing methods for artifact removal. This manuscript builds upon the work of Cooper et al. [91] and Huang et al. [15], which focused on artifacts induced by motion and interfering with NIRS monitoring. The review conducted by Huang et al. [15] was focused on mitigating fNIRS motion artifacts, and the review conducted by Cooper et al. [91] was conducted over a decade ago. As such, the information and level of detail presented in this review still fills a significant literature gap.

The results of this review indicate a lack of “gold-standard” for cerebral NIRS signal artifact management. There is vast disagreement regarding which method has superior functionality. It proved difficult to conduct an analytical comparison using quantified data due to the heterogeneity of data sources (simulated and experimental data), sampling rates used, types of artifacts that each method was able to remove, and differences in the signal characteristic of what an “artifact” was. Further, not having consistent “effectiveness” metrics and made it difficult to compare methods. Some articles compared the processed signal to the identified “true hemodynamic response function” and others were compared to manually cleaned signals. There were no clear differences in the types of artifact removal methods that were devised for one cerebral NIRS signal type (HbO and HHb, HbO only, rSO_2_, optical density), and there were no clearcut advantages or specific applications supporting the choice of one signal type over another.

There were four studies that presented artifact management methods that included the most robust comparisons of the effectiveness of their proposed methods to that of four or more using patient data [17,32,33,40]. Each of the methods were developed for artifact removal in HHb and HbO signals. However, there were many variables that made it difficult to compare the results between articles. The evaluation of the effectiveness of one method over the other did not have consistent metrics. Gao et al. used metrics of R and SNR [40], Jahani et al. used metrics of MSE, E_p_, R^2^, and AUC-ROC [33], and Fishburn et al. also used AUC-ROC, maximum activation t-statistic, and the greatest number of mesh vertices with positive significant values [32]. Despite the consistent use of AUC-ROC as an evaluation metric, the most successful methods remained difficult to compare, as AUC-ROC was calculated between the processed signal and the true HRF by Jahani et al. [32] and AUC-ROC was calculated between the processed signal and the motion-free signal by Fishburn et al. [33]. Effectiveness was also difficult to quantify due to the breadth of the types of artifacts and differing definitions of what the correct signal should look like. An example of this is illustrated by the fact that the Spline-SG method performs well in two of the comparison articles but has poor performance in another using peak activation [45,46,49]. The Fishburn et al. TDDR method seemingly performs much better in metrics of AUC-ROC for simulated data and is successful in experimental data using the t-statistic; however, the limitations of this method in removing high-frequency artifacts were noted and could be debilitating in the success of this method [33]. Robertson et al. also presented a comparison of a recursive least squares adaptive filter, wavelet-based method, multi-channel linear regression-based method, and an ICA-based method. The ICA-based and multi-channel linear regression-based methods performed the best when SNR was calculated; however, this comparison was conducted on only four subjects and did not include any hybrid methods of more complexity [17]. As such, its conclusions do not describe the leading cerebral NIRS artifact management methods. Similarly, the comparison conducted by Fishburn et al. [33] presents one of the most robust analyses of several methodologies that have been developed and were included in this review. The examination used a single dataset and serves as an example of the methodology that must be followed to elucidate a leading cerebral NIRS artifact management method. However, this methodology only compared the proposed TDDR method to six others, only one of which were hybrid methods.

Many methods utilized auxiliary data to detect artifacts either using accelerometer data for motion detection or comparing other physiological signals to detect the incidence of artifacts. The use of auxiliary data for cerebral NIRS artifact detection limits the widespread applicability of the method as it expands the requirements for the experimental setup.

The interconnected nature of brain hemodynamic monitoring makes it useful to leverage artifact removal methods that have been developed for other cerebral bio-signals. There are potential benefits to using ABP or ICP amongst other bio-signals typically recorded simultaneously in the ICU to develop a more robust artifact management method. However, the intricacies in the morphologies seen in artifacts in cerebral NIRS signals necessitates the validation of methods specifically on this signal type. Artifact management for cerebral NIRS signals must be able to address the unique artifact types that occur during ICU recording that may not occur in ICP or other cerebral hemodynamic monitoring techniques. These include, but are not limited to, artifacts originating from blood pooling beneath the NIRS optodes in incidences of hematoma, the effect of skin colour on the light emitted, and shifting of the NIRS optodes on the scalp due to adhesive failure.

The performance of the hybrid methods, particularly in the robust comparisons conducted by Gao et al. [40] and Jahani et al. [32], indicate the advantages of integrating multiple signal analysis techniques to remove the complex and varied morphological artifacts in cerebral NIRS data. This validates the claim that it is likely using a hybrid methodology that the most robust artifact management method will be constructed. The method presented by Gao et al. indicates that the three components of the hybrid method were each used to remove specific forms of artifacts [40], as was discussed in Section 3.1.6. The successes of the existing hybrid methods and the absence of any method developed that combines the utility of sequentially applied signal analysis techniques with machine learning indicates that there remains a significant knowledge gap in cerebral NIRS artifact management.

There were 15 articles that outlined methods that were focused solely on improving data quality through the reduction of signal baseline drift and physiological noise. However, only two methods had in-depth comparisons of proposed methods to other methods. Nguyen et al. and Santosa et al. both provided quantitative evidence that their proposed methods (RLSE and AR-IRLS, respectively) performed better than the methods to which they were compared [80,84]. Nguyen et al. used metrics of CNR and t-value compared to the true heart rate [80]. Santosa et al. quantified effectiveness using the AUC-ROC between the cleaned and processed signal [84]. The differences between these two studies, specifically using different metrics to quantify effectiveness, made it impossible to determine which method presented was the most effective. Due to the inability of these methods to remove more substantial motion artifacts, they will not be able to be applied in isolation to provide clean cerebral NIRS data streams, providing a further need for a hybrid artifact management method.

This review provides insights into the available artifact management methods for cerebral NIRS signals; however, comparisons between methodologies is only possible to conduct within articles due to the prevalence of limitations in this literature. To properly elucidate a leading method for cerebral NIRS artifact management, it requires that the limitations discussed in the proceeding section be adequately addressed.

### 4.1. Limitations of Literature

The most glaring limitation to the existing literature is the heterogeneity in the metrics used to quantify the effectiveness as has been discussed throughout. However, there are additional limitations to the literature that made the comparison of methodologies difficult.

There were several different signal types (HbO, HHb, tHb, rSO_2_, and optical density) and recording hardware types used to measure cerebral NIRS data from study participants. Each signal type had to be derived using different algorithms based on the measured absorbance of varying infrared wavelengths. This introduced variance in the signal morphologies on which models were constructed and tested. There were significant differences between studies in the instructions given to them. For healthy volunteers, some recordings were conducted while patients were asleep, still, or performing prescribed actions. This made the comparison of methods between articles ineffective due to the different artifacts potentially present depending on the activity of the volunteer. Additionally, some of the participant groups from which data were measured were healthy and others had active or past impaired neurological function. The variety of signal types, hardware types, actions performed by participants, and participant neuropathology introduced variance in the signal morphologies and artifacts observed in the cerebral NIRS signals between studies, which further introduced difficulty in comparing methods between studies. Despite that, this review examined time-series methods of identifying artifacts, and the number of variables to consider during recording emphasizes the need for robust external validation of methods. This could be accomplished using publicly available synthetic datasets with a diverse library of artifact types occurring across the datasets as well as corresponding cleaned signals. Through standardized comparison methods such as MSE, AUC-ROC, and computational time, a more objective depiction of the efficacy of artifact management methods could be elucidated. This methodology would accelerate the development of a “gold-standard” method as the weaknesses of each method could be more easily identified.

There were only three studies that indicated that the data used to construct models were sampled from patients with active or a history of neurological impairment [55,64,88]. The use of cerebral NIRS data is significant in evaluating impaired cerebral autoregulation, and as such, there is clearly a lack of research regarding artifact management methodologies that can address artifacts present in cerebral NIRS data during neuropathological states. Artifacts that occur during recordings, specifically when neuropathology is impaired, pose significant barriers to the widespread adoption of cerebral NIRS as an ICU hemodynamic monitoring tool. These artifacts include motion artifacts that result in large spikes in magnitude or substantial signal drift due to device disconnection. Both can result in false alarms that either unnecessarily distract ICU staff or put the patient at risk if something is missed.

### 4.2. Limitations of Review

There were several inherent limitations with this systematically conducted scoping review. This study only covers research articles published prior to 26 January 2024 and may not reflect the most recent research. The articles included were only those published in English, introducing a potential language bias. Due to the heterogeneity of the study types discussed above, there may have been inadequate conclusions that were able to be made due to a lack of sufficient information to conduct a meta-analysis.

### 4.3. Future Directions

Future directions based on these findings should include an updated study reviewing the effectiveness of each NIRS artifact removal method on the same data using the same metrics. It would be very beneficial to perform an in-depth comparison of several leading methods using the same dataset to truly evaluate the efficacy of one over the other. Different types of artifacts should be considered in this evaluation including rapid spikes in magnitude, signal disconnection artifacts, baseline drift, small oscillations, and artifacts resulting from physiological noise, similar to the study that was conducted by Fishburn et al. [33]. Limiting the number of variables to consider when investigating the effectiveness of each method will allow for a more robust analysis. The results of effectiveness will be able to be directly interpreted allowing for leading methods to truly be elucidated.

As was identified in this review, hybrid methods are able to identify varying types of artifacts using various integrated signal analysis methods. Future work that will be conducted in this lab will involve the development of a methodology that employs several signal analysis techniques, many of which were discussed in this review. This layered approach will involve the following:Threshold-based methods—will be used to detect extraneous data points as well as signal drift based on the expected magnitude of signals.Time-series autoregression-based methods—will be used to detect large magnitude spikes common in high frequency artifacts that occur during patient motion.Wavelet or Fourier transformation-based methods—the transformation of the time-series cerebral NIRS signals into the time-frequency or frequency domains will allow for the detection of artifacts in the oscillatory behavior of the cerebral NIRS data.Waveform morphology detection-based methods—a catalog will be developed based on high frequency data such that complex morphological artifacts can be identified using their morphological structure like those that have been based off the Hu et al. morphological clustering and analysis of ICP algorithm [92], which has been the basis for several morphology-based signal detection algorithms for ICP signals [93,94,95].

All these techniques will be applied using machine learning and will be merged into a single autonomous artifact management pipeline using machine learning techniques.

## 5. Conclusions

The results of the search conducted across multiple databases yielded cerebral NIRS artifact management of two types: (1) motion and disconnection artifact removal methods; (2) signal quality improvement and physiological/other noise removal methods. In conclusion, this review provides a comprehensive overview of the existing methods for artifact management in cerebral NIRS through an in-depth analysis of each method. However, the heterogeneity of the metrics used to quantify effectiveness, variations in the definitions of “artifacts”, and differences in the datasets used do not allow for a leading method in this application to be elucidated. There were two articles that presented hybrid methods and their efficacies relative to other methods in removing different artifact types. This further indicates that it is likely through a sequential layered approach that the most robust artifact removal can be conducted. Homogeneity in the metrics used to quantify effectiveness in each method and comparisons conducted on a single set of data would allow for a more accurate depiction of the efficacy of each method, and this was demonstrated in one article included in this review. Future work should involve the creation of a robust methodology to compare artifact management tools using a single dataset. The information presented will be foundational to the development of a sequential layered artifact management pipeline by our lab. It will make use of dynamic thresholding, time-series autoregressive models, and wavelet-based models, all of which were discussed in-depth in this review, and integrated using machine learning techniques to accurately remove artifacts from cerebral NIRS signals in real time.

## Figures and Tables

**Figure 1 bioengineering-11-00933-f001:**
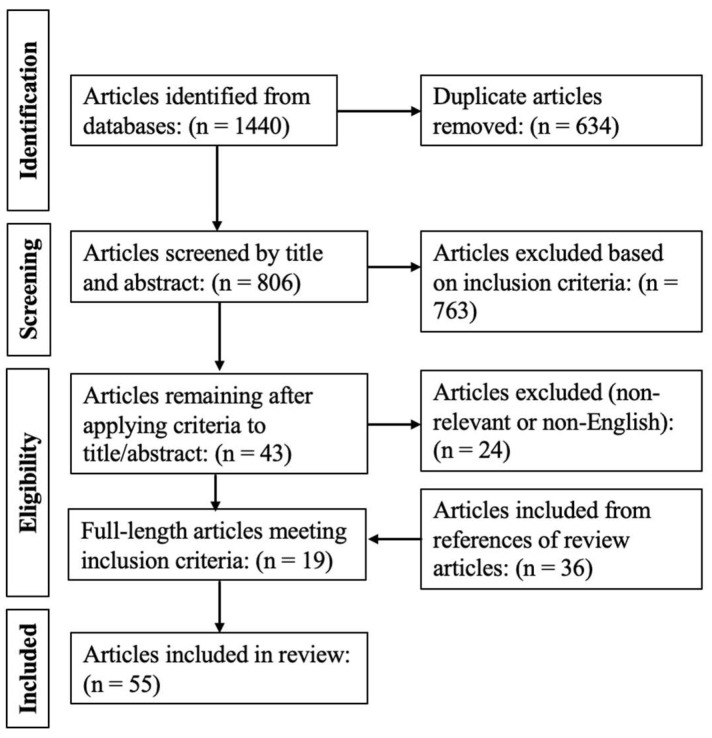
PRISMA flowchart of results for systematically conducted scoping review.

**Table 1 bioengineering-11-00933-t001:** Summary of motion and other disconnection artifact removal methods.

Artifact Removal Methods	Number of Studies Included	Number of Subjects	Signal Types (If Specified)	Effectiveness
Accelerometer-based	9	94	HHb and HbO	The t-test for significant and non-significant channels [49] indicated that there were no differences for non-significant channels in the performance of NoMC, UpMC, and HighMC methods for HbO and HHb signals and had no differences for HbO signals. However, the t-test for statistical significance indicated that there was a difference in the performance in HighMC compared to UpMC and NoMC in HHb significant signals [49]. In the analysis conducted by Metz et al., the sensitivity in identifying artifacts ranged from 86% to 96% in human scorers, was 92.2% using ABAMAR, 77.1% using MARA, and 94.2% using AMARA. However, AMARA did struggle with non-movement artifacts [50]. Siddequee et al.’s model that used a 3-axis accelerometer, gyroscope, and magnetometer to identify artifacts using inertial measurement unit (IMU) data and estimated artifacts using an autoregressive model with exogenous input (ARX) had the highest SNR (average of 15.38 dB) compared to when the ARX had input from just the accelerometer and gyroscope or just the accelerometer [36]. Sweeney et al. developed an adaptive filtering method that used the correlations between NIRS-based signals and accelerometer data to increase the SNR from −14.37 dB to 8.44 dB [31] Sweeney et al. also developed a method of artifact removal based on accelerometer data. There was no effectiveness metric provided; however it was stated that signal quality was improved using this method [51]. The agreement between the ABAMAR method and human observers on artifactual points was 79% with a 21% false positive rate [47]. Blasi et al. used accelerometer data to detect motion and adjust thresholds of an adaptive filter to remove artifacts. It achieved a ΔSNR of 0.535 dB [42]. Kim et al. presented an algorithm to detect artifacts using accelerometer data and remove artifacts using an adaptive filter; however, no comparison or quantitative data was provided regarding effectiveness [43] The BLISSA^2^RD algorithm decomposed fNIRS signals using ICA-ERBM [52] to temporally embed accelerometer data and used shared components to detect artifacts with CCA [53]. The ICA-ERBM algorithm was outperformed the fastICA algorithm [54]. Artifacts were estimated using ERBM source space [53]. This algorithm improved the SNR of continuous hemodynamic signals up to 10 dB and reduced motion artifacts by an order of two, outperforming several conventional methods in extracting the HRF [53]. This method outperformed an ICA-based, wavelet-based, and spline interpolation-based method in the metrics of RMSE and correlation in HbO signals; however, it did not outperform these methods in HHb signals [53].
Wavelet-based	8	307	rSO_2_, HHb and HbO, optical density	The Morlet wavelet based method developed by Bergmann et al. achieved a removal rate for simple artifacts of 100%, 99.8%, and 99.7% in HC, SP, and TBI datasets, respectively [55]. The kbWF-based method developed by Chiarelli et al. showed largest improvements in MSE (−24%) and ΔSNR (55%) compared to several other wavelet and PCA-based methods [56] In a comparison of several wavelet methods, the best performance was by WPD_fk4_ with the highest reduction in artifacts (−26.40%) and greatest ΔSNR (16.11 dB) of all single stage motion artifact correction techniques. The best performance in the two-stage for SNR was WPD_db1_-CCA with a ΔSNR of 16.55 dB, and the best for avg. motion artifact removal was WPD_fk8_-CCA with 41.40% [57]. Molavi and Dumont presented a TIWT-based method The mean artifact power attenuation between the two subjects was −17.26 dB, and the average NMSE was −13.99 dB [22]. In the Debauchies wavelet based method presented by the same authors, an average NMSE of −15.39 dB was achieved between three subjects [23]. Perpetuini et al. indicated that the presented Morse CWT-based method had the highest SNR (close to 5.5), lowest MSE (below 1), highest beta values (close to 0.9), and highest t-stat (close to 22) compared to the other wavelet-based, PCA-based, spine-based, and correlation-based methods to which it was compared [58]. The proposed DWT-based method presented by Wei et al. had a SNR value above 0 dB and an R^2^ value above 0.4. The wavelet filtering method to which it was compared had an SNR value below −10 dB and R^2^ value close to 0 [59]. Robertson et al. presented a discrete wavelet-based method that was able to achieve an average increase in SNR for NIRS signals when λ = 760 nm and λ = 830 nm of 5.96 dB and 4.93 dB, respectively. However, this method struggled compared to regression-based and ICA-based methods when the temporal location of the motion artifacts was not known [17].
Machine learning-based	3	50	HHb and HbO	There was a single method that leveraged machine learning techniques for artifact identification and removal. Feature engineering was used to try to identify artifacts based on the power density fraction, sample entropy, autocorrelation, and area under the curve for the signal based on artifact-free signals. The data was then fed into a Random Forest algorithm. However, there was limited success in artifact identification based on the engineering feature selection [37] Kim et al. presented a deep learning architecture capable of extracting features from fNIRS data and removing artifacts. This method performed better than an autoregressive model [60] and wavelet-MDL method [61] in limiting MSE and maximizing AUC-ROC in performance activation detection [62]. Lee et al. presented a BPNN with an AdaM optimizer [63] that used inputs from a 10-band wavelet transformation of fNIRS signals to achieve a CNR in corrected channels and in the channels in a region of interest activated by the subject walking of 0.63 and 0.73, respectively [64]. This method outperformed an HRF smoothing method [65], a wavelet denoising method and the wavelet-MDL method [61].
Filter-based	6	63	HHb and HbO	The comparison of adaptive, Wiener, and Kalman filters for slow, medium, and fast head movements yielded average ΔSNRs for each filter as 3.4396, 6.6980, and 7.6548, respectively [38,39]. An extension of Izzetoglu et al.’s Kalman filtering method [39] used an ARMA model to translate fNIRS signals into the state space and was able to increase the ΔSNR from 8.7 dB to 10.4 dB [44]. Dong and Jeong proposed an extended Kalman filter method that reduced RMSE and PRD by more than 40% in HbO and HHb signals compared to a linear Kalman filter, improved RMSE, PRD, and correlation coefficients between the recovered and true HRF had 34% increase in HbO and 62% in HHb compared to the linear Kalman filter [45]. Huang et al. presented a dual-stage median filter that was able to achieve the best average in SDR and NMSE with averages of 1.185 and 0.63, respectively [66], compared to the spline interpolation [41] and wavelet-based method [23]. Robertson et al. presented a recursive least squares-based adaptive filter that performed poorly in artifact removal compared to the SNRs achieved by wavelet-based, ICA-based, and regression-based methods [17].
Component analysis-based	4	40	HHb and HbO	The ICA-based method presented by Shi et al. resulted in an SNR increase of 2.336 and 2.139 in HHb and HbO signals, respectively, compared to an increase of 1.191 and 1.118, respectively, using a wavelet based method [67]. The tPCA-based method presented did not perform significantly better than the spline-based and wavelet-based methods by MSE and R^2^ when compared to true HRF; however, a significant increase was noted compared to when no correction was conducted [28]. The BLISSA^2^RD algorithm decomposed fNIRS signals using ICA-ERBM [52] to temporally embed accelerometer data and used shared components to detect artifacts with CCA [53]. The ICA-ERBM algorithm was outperformed the fastICA algorithm [54], as such, artifacts were estimated using ERBM source space [53]. This algorithm improved the SNR of continuous hemodynamic signals up to 10 dB and reduced motion artifacts by an order of two, outperforming several conventional methods in extracting the HRF [53]. This method outperformed an ICA-based, wavelet-based, and spline interpolation-based method in the metrics of RMSE and correlation in HbO signals; however, it did not outperform these methods in HHb signals [53]. Robertson et al. presented an ICA-based method that was able to achieve an average increase in SNR for NIRS signals when λ = 760 nm and λ = 830 nm of 5.62 dB and 2.76 dB, respectively. This method also achieved the best performance in SNR (3.20 dB, 3.67 dB) when the motion was unknown (compared to a wavelet-based, 30-channel regression-based and adaptive filter-based method) [17].
Hybrid	5	72	HHb and HbO, tHb	In the hybrid method proposed by Gao et al., the R value between the process and clean signal was close to 0.8 (none of the other methods are above 0.6) and the SNR between these two signals was above 0 (none of the other methods were above 0) [40]. Two datasets were used to analyze the hybrid method proposed by Jahani et al. There were conflicting results between the two datasets. The hybrid method was compared to four others (CBSI, tPCA, Spline-Rloess, wavelet filtering) and based on the metrics used (MSE, E_p_, R^2^, and AUC-ROC) [32]. More detail is shown in Appendix A. The hybrid method developed by Scholkmann et al. was not compared to anything. This method was evaluated by calculating PRD, RMSE, and R between the clean data and the data with induced motion artifacts with and without applying MARA. Employing the proposed MARA on three datasets of an undisclosed size and demographic resulted in an average change of −89.7% in PRD, −89.8% in RSME, and an increase of 61.6% in R [41]. Zhou et al. presented a method that used two moving standard deviation filters for artifact detection and spline interpolation as well as SG filtering to remove artifacts, achieving an SNR of 2.41 dB. It was compared to using spline interpolation and SG filtering only, which resulted in SNRs of −19.96 dB and −23.79 dB, respectively [46]. Gu et al. proposed a hybrid method that used thresholding for artifact detection, EMD for artifact removal, and spline interpolation to maintain signal continuity. It had an SNR increase of 53%, MSE reduction of 47%, and had an R^2^ value of 0.79 [27].
Other	9	147	HHb and HbO, optical density	The AR(P)-IRLS method proposed by Barker et al. was compared to three other OLS-based methods. The AUC-ROC values for the AR(P)-IRLS were consistently higher than all other compared metrics [60]. An extension of this method that involved the application of a Kalman estimator for online applications had similar performances to the AR-IRLS in specificity, sensitivity, and FPR in simulation results [34]. Cui et al. presented a correlation-based signal improvement algorithm that increased the CNR to 2.59 from 1.31 and 1.28 in HHb and HbO signals, respectively [24]. Fishburn et al. provided the most robust comparison of methods. They compared the proposed TDDR method to CSBI, MARA, tPCA, kurtosis wavelet, and Spline Savitzky–Golay filtering [33]. The TDDR method depicted better results than other methods based on AUC-ROC based on simulated data, the maximum activation t-statistic, and the greatest number of mesh vertices with positive significant (*p* < 0.05) values on real data. More detailed information is included in Appendix A. Wang and Seghouane presented a method that used discrete cosine transformation coefficients to estimate the signal and remove artifacts. The MSE between true resting state signal and restructured signal were 0.020876 for TARA, 0.015514 for TARA (non-convex) and 0.0037094 for the proposed method, indicating significant improvement [68]. Raggam et al. presented three methods as a part of a larger toolbox to reduce artifacts from fNIRS signals; however, there was no quantification provided for effectiveness [69]. Sutoko et al. used three signal characteristics (sudden increases in magnitude, shifting baseline magnitude, and intertrial correlation) to achieve a 71.8% rejection accuracy of artifact ridden signals compared to visual-based [70]. Robertson et al. presented a 30-channel regression model that was able to improve the SNR to 3.01 dB (λ = 760 nm) and 2.54 dB (λ = 830 nm) when the incidence of motion was unknown and 5.67 (λ = 760 nm) dB and 4.56 dB (λ = 830 nm) when it was known [17]. Sweeney et al. presented an EEMD-CCA method that was able to reduce motion artifacts from fNIRS signals, achieving an ΔSNR of 3.5 dB, 49.4% artifact reduction, and 0.68 correlation with the ground truth signal. This method outperformed an EEMD-ICA [71] and a wavelet-based method [35].

Where ABAMAR = Accelerometer-Based Automatic Motion Artifact Removal, AR-IRLS = Iterative Autoregressive Least Squares, ARMA = Autoregressive Moving Average, AUC-ROC = Area Under the Receiver Operating Characteristic Curve, AMARA = Automated Motion Artifact Removal Algorithm, ARX = Autoregressive Model with Exogenous Input, BLISSA^2^RD = Blind Source Separation and Accelerometer based, BPNN = Back Propagation Neural Network, Artifact Rejection and Detection, CCA = Canonical Correlation Analysis, CBSI = Correlation-Based Signal Improvement, CNR = Contrast-to-Noise Ratio, DWT = Discrete Wavelet Transform, EEMD-CCA = Ensemble Empirical Mode Decomposition with Canonical Correlation Analysis, EEMD-ICA = Ensemble Empirical Mode Decomposition with Independent Component Analysis, ERBM = Entropy Rate Bound Minimization, fk4 = Fejer–Korovkin (wavelet type) with 4 coefficients, fNIRS = Functional Near-Infrared Spectroscopy, FPR = False Positive Rate, HbO = Concentration of Oxyhemoglobin, HHb = Concentration of Deoxyhemoglobin, HC = Healthy Controls, ICA = Independent Component Analysis, IMU = Inertial Measurement Unit, kbWF = Kurtosis-Based Discrete Wavelet Filtering, MDL = Minimum Description Length, MARA = Motion Artifact Reduction Algorithm, MSE = Mean Squared Error, NMSE = Normalized Mean Squared Error, NoMC = No Motion Correction, PRD = Percent Root Difference, RMSE = Root Mean Squared Error, R^2^ = Coefficient of Determination, SDR = Signal Distortion Ratio, SNR = Signal-to-Noise Ratio, Spline-Rloess = Spline with Robust Locally Estimated Scatterplot Smoothing, tPCA = Temporal Principal Component Analysis, TARA = Targeted Artifact Removal Algorithm, TARA (Non-Convex) = Non-Convex Targeted Artifact Removal Algorithm, TDDR = Temporal Derivative Distribution Repair, TIWT = Translation Invariant Wavelet Transform, UpMC = Up Sampled Motion Correction, WPD = Wavelet Packet Decomposition.

**Table 2 bioengineering-11-00933-t002:** Summary of signal quality improvement and physiological/other noise removal methods.

Artifact Removal Methods	Number of Studies Included	Number of Subjects	Signal Types (if Specified)	Effectiveness
Signal drift removal	2	12	HHb and HbO	Shah and Seghouane presented a drift estimating method using wavelet thresholding that was able to achieve a CNR above 6 for HbO channels and above 5 for HHb channels [25]. Seghouane and Ferrari [78] conducted a study using the previous method that was able to outperform two established methods from a NIRS statistical parametric mapping toolbox [78,79] 9/18/2024 10:38:00 AM
Physiological and other noise artifact removal—NIRS only	8	71	HHb and HbO	The RLSE method presented by Nguyen et al. resulted in noise reduced in HbO and HHb by 77 and 99%, respectively, in CNR [80]. Zhang et al. proposed an adaptive filtering method that was able to reduce the effects of global interference in HbO signals but did not drastically improve HHb signals, both of which were measured in CNR [72]. Further analysis was conducted by this author on an adaptive filtering method, which indicated that 71% of HbO measurements exhibited an improvement in CNR [81]. Ortega-Martinez et al. proposed a multivariate Kalman filter tuned using tCCA and was able to demonstrate a 60% reduction in the RMSE compared to the use of a GLM [29]. Santosa et al. presented an ICA-based method for noise removal where the SNR of all HbO signals improved from 0.66 to 4.33. Based on the t-statistic, the ICA-based method performed better than low-pass filtering [82]. Chi et al. presented an EMD-based method that was able to estimate the heart rate within 80 to 90% accuracy [83] Santosa et al. conducted a robust analysis of several signal processing methods. The best performance in AUC-ROC across datasets was SS regression + ME + AR-IRLS method. SS regression + AR-IRLS method had similar success. Pre-processing did not have dramatic effect on AUC-ROC [84]. Guerrero-Mosquera et al. proposed the use of running correlation (global information) to detect noisy channels. It was compared to using cross-correlation (local information) [85]. Using local information resulted in an AUC-ROC increase of 91.23%, and the use of global correlations had an increase of 60.57% [85].
Physiological and other noise artifact removal—Auxiliary signals	5	51	HHb and HbO	Three methods proposed by Baruernfield included TF, CAR, and ICA methods. TF had the best performance in both HHb and HbO improvements of CNR for patient datasets. For TF, CNR improvements for HbO was within the range of 3.7% to 188.8% and for HHb from −0.6% to 55.0% [26]. The method proposed by Bontrager et al. used simultaneously recorded BP signals to reduce the correlations between fNIRS signals and BP [86]. The ICA-based method proposed by Kohno et al. was successful in removing artifacts resulting from blood skin flow based on correlation coefficients [87]. The coefficient of correlation was 0.724 between the identified component and changes in the skin blood flow for the first patient, and the coefficient of correlation was 0.789 for the second patient [87]. Sato et al. [88] presented a method of extracting scalp hemodynamics using PCA and removing noise using a GLM (ShortPCA GLM). It was compared to several methods and had the best performance based on metrics of R^2^ and specificity. The use of the standard GLM had a better performance in sensitivity. They were compared to fMRI data. Von Lühmann et al. developed a method using GLM with t CCA, which when compared to GLM with short separation, outperformed in metrics of HbO correlation, RMSE, F-score, p-value, and power spectral density [30].

Where AUC-ROC = Area Under the Curve—Receiver Operating Characteristic, AR-IRLS = Iterative Autoregressive Least Squares, BP = Blood Pressure, CAR = Common Average Reference, CNR = Contrast-to-Noise Ratio, EMD = Empirical Mode Decomposition, fMRI = Functional Magnetic Resonance Imaging, GLM = General Linear Model, HbO = Oxyhemoglobin, HHb = Deoxyhemoglobin, ICA = Independent Component Analysis, ME = Mixed Effect, PCA = Principal Component Analysis, RLSE = Robust Least Squares Estimation, RMSE = Root Mean Square Error, SNR = Signal-to-Noise Ratio, SS = Short Separation, tCCA = Targeted Canonical Correlation Analysis, TF = Transfer Function.

## Data Availability

Not Applicable.

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
