# Peer review of "Artifact Management for Cerebral Near-Infrared Spectroscopy Signals: A Systematic Scoping Review"

_bioengineering, 2024, doi:10.3390/bioengineering11090933_

Round 1

Reviewer 1 Report

Comments and Suggestions for Authors

Report on 'Artifact management for cerebral near-infrared spectroscopy signals: A systematic scoping review' by Bergmann et al. The authors reviewed methods proposed to clean the NIRS signals. They found two types of methods in the literature. 1. Motion and disconnection artifact removal method, 2) signal quality improvement and physiological/other noise removal method. The main conclusions are: 1) cross-comparison of the methods is impossible due to the heterogeneity of the methods used, and 2) the lack of a gold standard prevents performance assessment.

The manuscript is well-written overall. Below are a couple of questions that arose while reviewing it.

What is the motivation for doing this review? Are they planning on developing their algorithm or using the best algorithm to clean their NIRS signals?

Different NIRS devices use different algorithms to calculate the hemodynamic variables. This aspect is not discussed in the manuscript. This aspect may also preclude the cross-comparison of methods.

Reviewer 2 Report

Comments and Suggestions for Authors

This paper presents a systematic review-based study titled, “Artifact Management for Cerebral Near-Infrared Spectroscopy Signals: A Systematic Scoping Review”. The topic is good however I have some points that need to be addressed given as under.

My detailed concerns on the study are:

1.      What are the key findings/insights of this review with reference to existing effective artifact management methods?

2.      Discuss the research gap in the literature which this systematic review aims to address.

3.      Highlight the key questions or hypotheses, if any.

4.      What is new in this systematic review that the existing systematic review studies do not have?

5.      The authors summarized various techniques in tabular form in the Appendix C which created an unnecessary length of the paper. It is suggested to discuss these techniques in the text making a section and subsections for various relevant techniques, the information about the algorithm, extracted features, dimension of features vector classifier, and detection accuracy may be presented. This will also increase the readability of the paper.

6.      In the appendixes the databases are discussed. However, the detailed information about databases publically available should be discussed in section 2, which is missing.

7.      In section 4.1, the future research direction should also include key issues that need to be addressed in the future research, this will be helpful for the future researcher.

8.      In the conclusion section, a clear summary of the findings of the review with broader significance must be provided. 

Reviewer 3 Report

Comments and Suggestions for Authors

The aim of this review was to thoroughly analyse the current research on artifact management techniques for cerebral NIRS signals recorded in both animals and humans. A methodical examination was performed across five databases following the guidelines of Preferred Reporting Items for Systematic Reviews and Meta-Analysis. A total of 806 distinct outcomes were obtained from the search. A total of 19 articles were included in this review, based on the specified inclusion/exclusion criteria. An additional 20 articles were identified in the references of select articles and were also included. The articles classified the methods into two main categories: (1) techniques for removing motion and other disconnection artifacts, and (2) methods for improving data quality and filtering physiological/other noise artefacts. These were further classified based on the type of method used. The topic is of interest, however, some comments need to be addressed.

In the abstract, what differs this review from previous reviews for artifact management.

.

What consequences for both research and clinical practice emerge from the absence of a gold standard for artifact handling?

Explain the manner in which the variation of evaluation criteria could be addressed to support more strong assessments and comparisond of artefact management techniques

What consequences result from the differences in the composition of artefacts in cerebral NIRS compared to other biosignals?

What effects on the accuracy and dependability of cerebral NIRS readings do various forms of artefacts have?

How may the whole effectiveness of cerebral NIRS systems be enhanced by combining several artefact control strategies?

Please highlight aims and objectives as well as the novelty of this review.

Please add a paragraph describing the paper’s organization by the end of the introduction.

What were the keywords used for searching?

There is a problem with paper numbering. Please revise.

Which are the main determinants of the suitable artifact management technique for a particular clinical use of cerebral NIRS?

In what ways might the field progress with the help of consistent evaluation measures for artefact management?

It is very weird that the appendix size is much bigger than the main document size. I recommend moving some parts to the main document.

Comments on the Quality of English Language

some grammar mistakes were detected

Round 2

Reviewer 2 Report

Comments and Suggestions for Authors

The authors have sufficiently addressed my point raised in the review-1.  I have no more comments.

Author Response

The reviewer indicated that there were no further comments. Thank you. 

Reviewer 3 Report

Comments and Suggestions for Authors

Thank you for addressing most of my comments but could you please address comment 12 as it was missed during revision?
